# Model Adaptive Tooth Segmentation

**Ruizhe Chen**[1]                                    RUIZHEC.21@INTL.ZJU.EDU.CN
**Jianfei Yang**[2]                                   FENGYANG@ANGELALIGN.COM
**Yang Feng**[3]                                      YANG0478@E.NTU.EDU.SG
**Jin Hao**[4]                                        JINHAOSCRB@GMAIL.COM
**Zuozhu Liu**[1]                                     ZUOZHULIU@INTL.ZJU.EDU.CN

[1] *Zhejiang University*

[2] *Nanyang Technological University*

[3] *Angelalign Tech.*

[4] *Stanford University*

**Editors:** Accepted for publication at MIDL 2023

## Abstract

Automatic 3-dimensional tooth segmentation on intraoral scans (IOS) plays a pivotal role in computer-aided orthodontic treatments. In practice, deploying existing well-trained models to different medical centers suffers from two main problems: (1) the data distribution shifts between existing and new centers, (2) the data in the existing center is usually not allowed to share while annotating additional data in the new center is time-consuming and expensive. In this paper, we propose a Model Adaptive Tooth Segmentation (MATS) framework to alleviate these issues. Taking the trained model from a source center as input, MATS adapts it to different target centers without data transmission or additional annotations, as inspired by the source data-free domain adaptation (SFDA) paradigm. The model adaptation in MATS is realized by a tooth-level feature prototype learning module, a progressive pseudo-labeling module and a tooth-prior regularized information maximization loss. Experiments on a dataset with tooth abnormalities and a real-world cross-center dataset show that MATS can consistently surpass existing baselines. The effectiveness is further verified with extensive ablation studies and statistical analysis, demonstrating its applicability for privacy-preserving tooth segmentation in real-world digital dentistry.

**Keywords:** Tooth Segmentation, Domain Adaptation, Cross-center Validation

## 1. Introduction

Automatic and accurate tooth segmentation on 3-dimensional (3D) intraoral scanned dental models is an indispensable prerequisite for computer-aided orthodontic treatments. Formally, an intraoral scan (IOS) is a 3D mesh surface that provides a digital impression of the tooth anatomy with 100,000 to 400,000 high-resolution (0.008-0.02mm) triangular mesh faces, while the 3D tooth segmentation is designated to classify each face to the gingiva and different teeth following the Federation Dentaire Internationale (FDI) standard (Herrmann, 1967). A plethora of works (Xu et al., 2018b; Hao et al., 2022; Lian et al., 2020; Zhang et al., 2021; He et al., 2021; Qiu et al., 2022; Zanjani et al., 2021) have been done with outstanding promising performance for automatic 3D tooth segmentation, aiming to reduce the expensive and time-consuming annotation cost by human experts, i.e., manual labor of

15 to 30 minutes to annotate a half-jaw. They design domain-specific deep neural networks that consume the raw mesh or simplified point clouds from IOSs for tooth segmentation. However, most of these methods require a large amount of annotated IOSs to achieve satisfactory results, while bearing the limitations regarding privacy and accuracy when directly applied to various medical centers with complicated cases in practice (Cui et al., 2021; Tian et al., 2019; Xu et al., 2018a).

The main challenges are illustrated in Fig. 1. On the one hand, different medical centers (large hospitals or small clinics) usually acquire IOSs from patients with diverse oral diseases, i.e., a certain disease that occurs frequently in one center is not necessarily ordinary in another. Such a data distribution difference might cause huge performance degradation when directly deploying the model trained on one center to another new center. On the other hand, though fine-tuning pre-trained models leveraging additional labeled samples in the new center or existing annotations from other centers could help improve the segmentation performance, the annotation process is labor-intensive and expensive, while annotations from other centers are usually not available either due to privacy or regulation issues. *In consequence, a new solution that can alleviate the data distribution gap to avoid performance degradation and, in the meanwhile, does not require additional labeled IOSs or data exchange among medical centers, is of high necessity to deploy existing models across different medical centers.*

Various unsupervised domain adaptation (UDA) methods have been proposed to alleviate the distribution gap between the source domain (existing centers) and target domain (new centers) without additional labeled data via adversarial learning (Tzeng et al., 2017; Zhang et al., 2018; Zou et al., 2018), statistic-based methods (Long et al., 2015; Sun and Saenko, 2016) and semantic alignment (Pan et al., 2019; Xie et al., 2018; Luo et al., 2019) etc. However, they require access to the source and target data simultaneously, which may be inapplicable due to privacy concerns. Against this backdrop, the source data-free domain adaptation (SFDA) paradigm provides a promising alternative solution, which addresses the UDA problem without the source data (Qiu et al., 2021; Stan and Rostami, 2021; Yeh et al., 2021; Kundu et al., 2021; Liu et al., 2021a; Bateson et al., 2021; Kothandaraman et al., 2021; Yang et al., 2022). Test-Time Adaptation enables source data-free adaptation of an existing model to online target data, which is detailedly discussed in Related Works in Appendix H. However, to our knowledge, few of these works investigate source-free 3D segmentation in meshes/point clouds, while their performance is yet unconsidered or might be unsatisfactory.

In this paper, we propose a privacy-preserving cross-center Model Adaptive Tooth Segmentation (MATS) framework to address the aforementioned challenges. Similar to SFDA, MATS only needs access to the trained source models for privacy-preserving adaptation across different medical centers, without the requirement for additional locally labeled IOSs data. Fully leveraging the characteristics of the tooth, MATS is designed to perform source-free 3D tooth adaptation in three aspects. Firstly, we propose a tooth-level feature prototype learning module to exploit the spatial relationship and local geometric features on each tooth for consistent knowledge transfer. We further employ a progressive pseudo-labeling strategy that encourages the source model to better adapt to different target centers. Lastly, a tooth-ratio prior corresponding to surface areas among different teeth and the gingiva is utilized to maximize the mutual information during domain adaptation for tooth segmentation.

Our experiments are conducted on two self-collected datasets: the first dataset is established with respect to dental abnormalities (i.e., abnormal tooth eruption, dentural diastema, and tooth defects) which simulates the largest domain distribution difference across medical centers in tooth segmentation; the second dataset is collected from 4 wild medical centers, which is in full accord to the real-world cross-center scenario. Experiments on two datasets demonstrate that MATS can surpass existing UDA and SFDA methods with significant and consistent improvements. Further statistical analysis and ablation studies exhibit the effectiveness of each component in MATS, as well as the advantages and limitations. Our work corroborates the great potential of state-of-the-art privacy-preserving deep learning solutions for future digital dentistry.

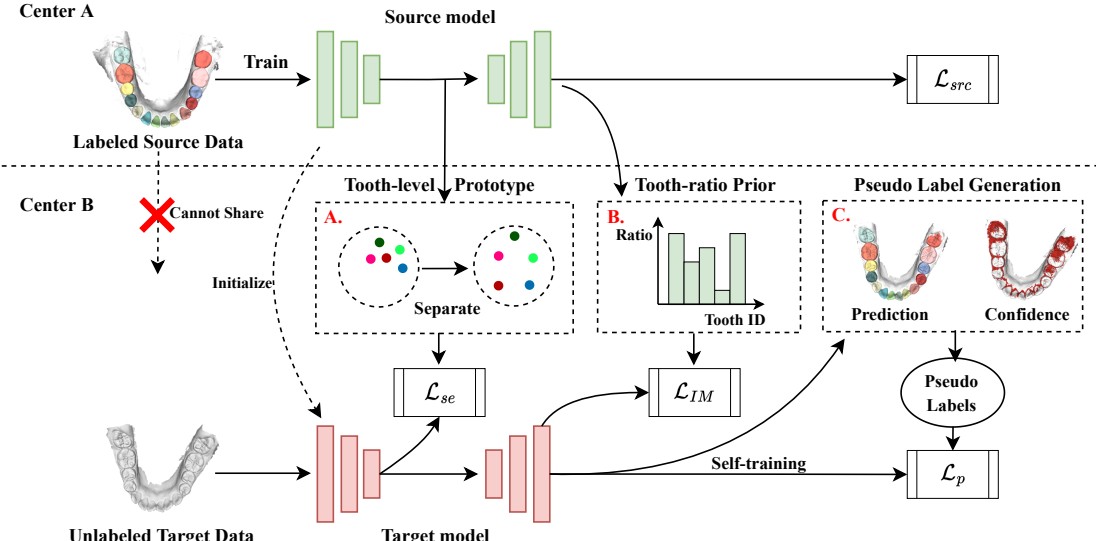

Figure 1: The framework of our MATS.

## 2. Methodology

The overall pipeline of MATS is illustrated in Fig. 1. It consists of two steps. First, a source model $f_s : \mathcal{X}_s \to \mathcal{Y}_s$ is trained with $N_s$ annotated samples $\{x_s^{(n)}, y_s^{(n)}\}_{n=1}^{N_s} \in \{\mathcal{X}_s, \mathcal{Y}_s\}$ in a source center (source domain) $\mathcal{D}_s$. Afterwards, we adapt the source model to the target center (target domain) $\mathcal{D}_t$ with $N_t$ unlabeled data $\{x_t^{(n)}\}_{n=1}^{N_t} \in \mathcal{X}_t$ while the source data is not available due to privacy concerns. Concretely, we learn the target model $f_t : \mathcal{X}_t \to \mathcal{Y}_t$ to generate segmentation $\{\hat{y}_t^{(n)}\}_{n=1}^{N_t}$ in the target center, with only $\{x_t^{(n)}\}_{n=1}^{N_t}$ and the source model $f_s$. Note that $\{x_s^{(n)}\}_{n=1}^{N_s}$ and $\{x_t^{(n)}\}_{n=1}^{N_t}$ denote down-sampled point clouds from the IOSs. MATS includes three novel designs: a tooth-level feature prototypes generation and alignment module, a progressive pseudo-label generation module, and an information maximization loss with tooth-ratio priors. These modules inherit the merits of prevailing domain alignment techniques while revamping them to perform robust tooth segmentation.

### 2.1. Source Model Generation

We train the source model $f_s : \mathcal{X}_s \to \mathcal{Y}_s$ with a supervised cross-entropy loss:

$$\mathcal{L}_{src}(f_s; \mathcal{X}_s, \mathcal{Y}_s) = -\mathbb{E}_{(x_s,y_s)\in(\mathcal{X}_s,\mathcal{Y}_s)} \sum_{i=1}^{n_s} \sum_{k=1}^{K} q_k^i \log \delta_k(f_s(x_s^i)), \tag{1}$$

where $\delta_k(v)$ denotes the $k$-th element in the softmax prediction, and $q^i$ is a one-hot 33-dimensional ground truth vector for the $i$-th point with $q_k^i = 1$ for tooth label $k$ and $q_k^i = 0$ otherwise. $K = 33$ denotes the 32 permanent teeth and the gingiva in FDI.

### 2.2. Tooth-level Feature Prototypes Generation and Alignment

In the tooth segmentation task, the appearance, location and frequency for each tooth category intend to remain consistent across patients from source or target centers. Therefore, we consider tooth-level features to be transferable across domains. Though source data is unavailable, we can instead retain tooth-level feature prototypes from the source domain. The generation operation of each feature prototype is as follows:

$$\mathcal{P}_k = \mathbb{E}_{x_s \in \mathcal{X}_s} \frac{\sum_i g_s(x_s^i) \mathbb{1}[y_s^i = k]}{\sum_i \mathbb{1}[y_s^i = k]}, \tag{2}$$

where $\mathcal{P}_k$ denotes the feature prototype for $k$-th tooth class, $\mathbb{1}[\cdot]$ is the indicator function, and $g_s(x_s^i)$ is the feature corresponding to the $i$-th point. It's worth noting that we only share the mean of features across source cases, which will not reveal information about dental cases. In the target domain, we measure the similarity between its features and generated prototypes for each point with the cosine similarity: $\phi(g_t(x_t^i), \mathcal{P}_k) = \frac{g_t(x_t^i) \cdot \mathcal{P}_k}{||g_t(x_t^i)|| \ ||\mathcal{P}_k||}$. For domain alignment, we expect the features of each point to be sufficiently similar to the closest prototype while away from the other prototypes, which means the similarity vectors should be close to one-hot encodings. Thus, we employ a self-entropy loss on similarity vectors: $\mathcal{L}_{se}(f_t, \mathcal{X}_t, \mathcal{P}_k) = -\mathbb{E}_{x_t \in \mathcal{X}_t} \sum_{i=1}^{n_t} \sum_{k=1}^{K} \phi(g_t(x_t^i), \mathcal{P}_k) \log \phi(g_t(x_t^i), \mathcal{P}_k)$. Moreover, we propose to add a softmax layer with low temperature $\mathcal{T}$ to further separate the prototypes $\mathcal{P}_k$ generated in Eq. (2) that $\mathcal{P}_k' = \frac{\exp(\mathcal{P}_k/\mathcal{T})}{\sum_k^K \exp(\mathcal{P}_k/\mathcal{T})}$. Correspondingly, the definition of $\mathcal{L}_{se}$ is changed as $\mathcal{L}_{se}(f_t; \mathcal{X}_t, \mathcal{P}_k')$. Detailed discussion on the effectiveness of $\mathcal{T}$ can be found in the Appendix B.

### 2.3. Progressive Pseudo Label Self-training

In our preliminary experiments, we have observed that the source model tends to be more confident and accurate in predictions on areas similar to the source domain while tending to make unconfident and wrong predictions on abnormal areas. Based on this, we propose to evaluate the prediction with its confidence level and assign pseudo-labels to points of high confidence. We propose a progressive self-training strategy. In each training step, We generate pseudo labels for only high confident points for self-training while excluding other points. Specifically, if the confidence level $\delta_k(f_t(x_t^i))$ that the $i$-th point belongs to the $k$-th class is higher than the threshold $\tau$, the pseudo label of point $i$ is assigned as $k$. The point

is excluded if $\delta_k(f_t(x_t^i)) < \tau, \forall k \in K$. The generation of pseudo-labels is online. Hence, as the model becomes more adapted to the target domain, more points of high confidence will be included during training. We adopt a cross-entropy loss on generated pseudo labels for self-training: $\mathcal{L}_p(f_t; \mathcal{X}_t, \tilde{\mathcal{Y}}_t) = -\sum_{(x_t, \tilde{y}_t) \in (\mathcal{X}_t, \tilde{\mathcal{Y}}_t)} \sum_{i=1}^{n_t} \tilde{y}_t^i \log \delta(f_t(x_t^i))$.

It is likely that the model might generate wrong labels with high confidence (Chen et al., 2019a). To alleviate this issue, we further evaluate the quality of pseudo labels by the similarity between their features and the corresponding prototypes. Pseudo labels with high similarity would be assigned larger weights. The re-weighted loss function is defined as:

$$\mathcal{L}_p(f_t; \mathcal{X}_t, \tilde{\mathcal{Y}}_t) = - \sum_{(x_t, \tilde{y}_t) \in (\mathcal{X}_t, \tilde{\mathcal{Y}}_t)} \sum_{i=1}^{n_t} \phi(g_t(x_t^i), \mathcal{P}_k^{'})) \tilde{y}_t^i \log \delta(f_t(x_t^i)), \tag{3}$$

where $\phi$ is the similarity between $g_t(x_t^i)$ and $\mathcal{P}_k^{'}$.

### 2.4. Information Maximization with Tooth-ratio Priors

In a universal dental model, the ratios between the teeth (and gingiva) tend to remain constant. For example, the number of points belonging to molars is larger than that of the incisors. We propose to preserve these ratios in the source domain as prior knowledge (Bateson et al., 2021) and combine it in Information Maximization loss $\mathcal{L}_{IM}$ (Peng et al., 2020; Liang et al., 2020). Detailed discussion on the consistency of tooth-ratio prior can be referred to Appendix A.3. The generation of tooth-ratio priors $\mathcal{R}_s$: $\mathcal{R}_s[k] = \frac{1}{N_s} \sum_{(x_s, y_s) \in (\mathcal{X}_s, \mathcal{Y}_s)} \frac{1}{|n_s|} \sum_{i=1}^{n_s} \mathbb{1}[y_s^i = k]$, where $\mathbb{1}[y_s^i = k]$ is the indicator function. The revised $\mathcal{L}_{IM}$ consists of (1) an entropy loss $\mathcal{L}_{ent}(f_t; \mathcal{X}_t) = -\mathbb{E}_{x_t \in \mathcal{X}_t} \sum_{i=1}^{n_t} \delta(f_t(x_t^i)) \log \delta(f_t(x_t^i))$ to encourage confident prediction where $\delta(f_t(x_t^i))$ is the K-dimensional prediction of the $i$-th point in target dental model $x_t$ after softmax; (2) a revised diversity loss $\mathcal{L}_{div}(f_t; \mathcal{X}_t) = D_{KL}(\hat{p}, \mathcal{R}_s)$ to encourage the accordance of global prediction and $\mathcal{R}_s$, where $D_{KL}$ is the Kullback–Leibler divergence, and $\hat{p} = \mathbb{E}_{x_t \in \mathcal{X}_t}[\delta(f_t(x_t))]$ is the mean output embedding of the entire target domain. The proposed information maximization $\mathcal{L}_{IM}$ with tooth-ratio priors $\mathcal{R}_s$ is written as: $\mathcal{L}_{IM}(f_t; \mathcal{X}_t) = \mathcal{L}_{ent}(f_t; \mathcal{X}_t) + \alpha \mathcal{L}_{div}(f_t; \mathcal{X}_t)$, where $\alpha$ is a weighted hyper-parameter. The final loss function for MATS is defined as $\mathcal{L} = \mathcal{L}_{IM} + \beta \mathcal{L}_{se} + \gamma \mathcal{L}_p$, where $\beta$ and $\gamma$ are weighting coefficient hyper-parameters. Detailed information on our method can be referred to in Appendix A.

## 3. Experiments and Discussion

### 3.1. Experimental Setup

**Datasets.** To comprehensively evaluate our method in the real world, we collected two datasets according to different clinical demands. The first dataset **AbnTeeth** consists of 2,649 3D IOSs that are manually segmented by professional dentists. In addition, labels of three dental abnormalities are annotated: (1) **Abnormal Teeth Eruption (ATE)** (2) **Dentural Diastema (DD)** (3) **Teeth Defects (TD)** (Proffit et al., 2006). We regard the normal dental models as the source domain, and the abnormal dental models as the

target domain, which simulates the greatest domain shift and forms three transfer tasks in tooth segmentation: ATE, DD and TD (normal → abnormal). The splits and detailed description are presented in Table 5 in the Appendix B. The second dataset **CrossTeeth** consists of 2,489 3D IOSs collected from 4 real-world clinical centers in different cities. In real world centers, distribution differences could be caused by different 3D scanning machines, demographic characteristics and patients. For instance, in a dental center that targets children and young patients, around 95% of patients have erupted teeth. Statistical analysis on the data distribution in different centers can be referred to Appendix B.

| Models | $\mathcal{D}_s$ | DD | | | TD | | | ATE | | |
|---|---|---|---|---|---|---|---|---|---|---|
| | | mIoU | DSC | Acc | mIoU | DSC | Acc | mIoU | DSC | Acc |
| Source only | ✓ | 83.76 | 88.00 | 94.04 | 80.38 | 86.02 | 91.62 | 83.60 | 88.32 | 93.58 |
| DAN | ✓ | 78.65 | 82.75 | 90.95 | 80.04 | 85.08 | 90.73 | 80.74 | 85.15 | 91.11 |
| PointDAN | ✓ | 84.05 | 87.79 | 94.11 | 85.02 | 90.17 | 93.57 | 82.25 | 86.46 | 92.07 |
| MSL | ✓ | 86.65 | 89.93 | 95.02 | 85.48 | 89.83 | 94.41 | 86.46 | 91.30 | 94.89 |
| AdaptSegNet | ✓ | 85.99 | 89.36 | 95.23 | 86.57 | 90.41 | 94.67 | 81.41 | 86.74 | 92.41 |
| SHOT | | 83.66 | 87.63 | 93.60 | 85.55 | 89.48 | 94.00 | 87.30 | 91.02 | 94.98 |
| AdaMI | | 84.31 | 87.92 | 94.63 | 83.52 | 88.42 | 93.20 | 87.00 | 90.86 | 94.83 |
| SS-SFDA | | 86.36 | 89.82 | 95.01 | 86.11 | 90.38 | 94.04 | 85.38 | 89.67 | 94.06 |
| Ours | | **87.90** | **90.99** | **95.38** | **87.32** | **91.22** | **94.61** | **87.32** | **91.10** | **94.98** |
| Oracle | | 91.38 | 94.36 | 96.26 | 90.46 | 93.26 | 95.84 | 92.32 | 94.82 | 96.88 |

Table 1: Results on the abnormal teeth dataset. Results are reported in percentage.

**Baseline Methods and Metrics.** To the best of our knowledge, there is no off-the-shelf SFDA architecture for the point cloud. Hence, we compare our method with state-of-the-art domain adaptation methods (i.e., DAN (Long et al., 2015), MSL (Chen et al., 2019b), AdaptSegNet (Tsai et al., 2018a)), point cloud domain adaptation methods (i.e., PointDAN (Qin et al., 2019)) and SFDA methods (i.e., SHOT (Liang et al., 2020), AdaMI (Bateson et al., 2021), SS-SFDA (Kothandaraman et al., 2021)) based on the same feature extractor (DC-Net). SFDA segmentation architectures (Yang et al., 2022; Liu et al., 2021a) are not compared because their designs (Fourier transformation, generator) on 2D images cannot be applied to point clouds. "Source only" and "Oracle" represent the performance of models trained with only labeled source data and labeled target data, respectively. We evaluate the performance with three metrics: mean Intersection-over-Union (**mIoU**), point-wise accuracy (**Acc**), and Dice Similarity Coefficient (**DSC**).

| Models | $1 \to 2$ | $1 \to 3$ | $1 \to 4$ | $2 \to 1$ | $2 \to 3$ | $2 \to 4$ | $3 \to 1$ | $3 \to 2$ | $3 \to 4$ | $4 \to 1$ | $4 \to 2$ | $4 \to 3$ | Avg. |
|---|---|---|---|---|---|---|---|---|---|---|---|---|---|
| Source only | 81.37 | 84.07 | 71.46 | 72.67 | 73.99 | 72.49 | 78.31 | 80.12 | 79.75 | 70.99 | 76.74 | 75.65 | 76.47 |
| SHOT | 80.38 | 84.16 | 72.57 | 77.27 | 79.9 | 78.74 | 82.77 | 85.45 | 81.3 | 76.12 | 79.56 | 79.12 | 79.78 |
| AdaMI | 81.32 | 84.01 | 71.89 | 76.49 | 81.36 | 78.99 | 81.27 | 82.01 | 81.04 | **79.81** | 78.3 | 79.35 | 79.65 |
| SS-SFDA | 81.02 | 85.11 | 73.27 | 77.89 | 80.01 | 79.76 | 83.21 | **86.55** | 82.01 | 78.61 | 79.81 | 79.63 | 80.49 |
| Ours | **83.12** | **85.56** | **76.3** | **80.14** | **82.94** | **83.03** | **83.59** | 86.07 | **83.17** | 79.13 | **81.05** | **82.12** | **82.19** |
| Oracle | 86.68 | 86.68 | 86.68 | 88.47 | 88.47 | 88.47 | 88.21 | 88.21 | 88.21 | 85.04 | 85.04 | 85.04 | 87.1 |

Table 2: Results (mIoU) on the cross-center dataset. "1", "2", "3", "4" represent 4 centers.

### 3.2. Ablation Study

We investigate the effectiveness of $\mathcal{L}_{IM}, \mathcal{L}_{se}$ and $\mathcal{L}_p$ on the DD transfer task. The results are shown in Table 2. When the three losses are applied separately, $\mathcal{L}_{IM}$ brings some improvement, while the other two suffer great performance drops. Though the improvement of $\mathcal{L}_{IM}$ is marginal, it plays an important role in enforcing the model to comply with cluster assumption. Applied with $\mathcal{L}_{IM}$, $\mathcal{L}_{se}$ and $\mathcal{L}_p$ can bring significant performance improvements, indicating that these two losses can perform effective alignment and purify the pseudo labels iteratively. Furthermore, to validate the mechanism of prototype separation of $\mathcal{P}'_k$ in $\mathcal{L}_{se}$, we visualize the cosine similarity matrix of prototypes before (left) and after separation operation (right) in Fig. 3. It can be observed that after the separation, similarities between different prototypes, especially adjacent prototypes decrease. To verify the effectiveness of our proposed tooth-ratio prior, we compared the performance of our method with vanilla $\mathcal{L}_{IM}$, which replaces the tooth-ratio prior with a uniform vector $\mathbf{1}_K$. As reported in Line 8 and 9 of Table 2, our proposed prior increases mIoU and DSC by 1.32% and 1.24%, respectively. Ablation studies on the hyperparameters $\alpha, \beta, \gamma$, threshold $\tau$ and temperature $\mathcal{T}$ can be referred to in Appendix D.

### 3.3. Overall Evaluation

We compare our method with different baseline methods on three subsets of tooth abnormality, with results presented in Table 1. Experiments have shown that our MATS outperforms all DA and Source-free DA methods on all datasets. The results illustrate that our tooth prior knowledge-based algorithm is more suitable for tooth segmentation tasks. It is noteworthy that our method outperforms the second best method by 1.25% and 0.75% improvement on DD and TD tasks. However, it is only a 0.02% improvement over SHOT on ATE. It may indicate that the distribution differences caused by the abnormal teeth eruption are comparably small, and therefore, multiple methods show better performance on this dataset. However, our method achieves a 4.24% improvement over SHOT on DD, indicating that our method is more generalized over different tooth segmentation tasks. We also provided the overall segmentation results of our method on cross-center dataset in Table 2. It can be observed that in real-world datasets, our methods outperformed the SFDA baselines in 10 of 12 adaptation tasks. our methods accomplished 2.42%, 2.54%, and 1.7% performance gain in terms of Average mIoU compared to the three baseline methods. Results on the two datasets both prove the superiority of our proposed method. We also provide comparison results on a public dataset (Achituve et al., 2021) to verify the generalization ability of our proposed framework in Appendix E. We also investigate the performance of our method on limited source data in Appendix F.

### 3.4. Tooth-level quantitative performance analysis

Our overall evaluation has shown that our method outperforms all baselines on the whole dataset. After that, we further compare the transfer ability on abnormal areas of our method with baselines. The adaptation results in terms of tooth category are reported in Table 3. Experiments have shown that our method outperforms the other methods in five out of seven categories, illustrating our superiority in dealing with teeth abnormalities.

| $f_s$ | $\mathcal{L}_{IM}$ | $\mathcal{L}_{se}$ | $\mathcal{L}_p$ | mIoU (%) | DSC (%) | Acc (%) |
|---|---|---|---|---|---|---|
| ✓ | | | | 83.76 | 88.00 | 94.04 |
| ✓ | ✓ | | | 84.31 | 87.92 | 94.63 |
| ✓ | | ✓ | | 45.47 | 56.82 | 73.73 |
| ✓ | | | ✓ | 11.93 | 15.93 | 21.43 |
| ✓ | ✓ | ✓ | | 87.53 | 90.67 | 95.39 |
| ✓ | ✓ | | ✓ | 86.63 | 89.99 | 95.05 |
| ✓ | ✓ | ✓ | * | 86.14 | 90.30 | 94.03 |
| ✓ | △ | ✓ | ✓ | 86.58 | 89.75 | 95.33 |
| ✓ | ✓ | ✓ | ✓ | 87.90 | 90.99 | 95.38 |

Figure 2: Ablation study of the losses. The "△" represents $\mathcal{L}_{IM}$ without tooth-ratio prior. The "*" represents unweighted $\mathcal{L}_p$.

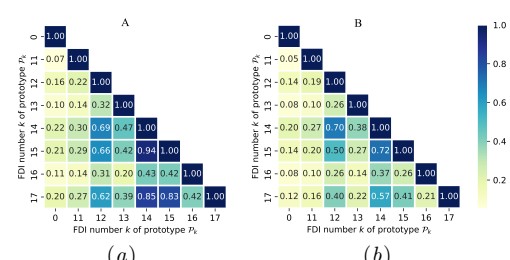

Figure 3: Cosine similarity between different prototypes (0, 11-17).

| Models | $\mathcal{D}_s$ | c-incisor | incisor | canine | $1^{st}$p-molar | $2^{nd}$p-molar | $1^{st}$molar | $2^{nd}$molar |
|---|---|---|---|---|---|---|---|---|
| Source only | ✓ | 88.14 (1.65) | 88.03 (1.89) | 91.20 (1.74) | 91.43 (3.88) | 90.86 (3.19) | 91.17 (2.82) | 80.52 (5.68) |
| DAN | ✓ | 88.15 (2.57) | 88.15 (2.97) | 91.56 (2.27) | 92.09 (4.52) | 90.69 (4.06) | 88.38 (4.65) | 64.97 (9.13) |
| PointDAN | ✓ | 86.92 (2.79) | 87.92 (2.86) | 90.84 (2.21) | 90.74 (3.56) | 87.11 (4.43) | 86.01 (4.49) | 61.09 (7.50) |
| MSL | ✓ | **90.62 (2.01)** | 90.07 (2.42) | 92.29 (1.86) | 92.77 (3.58) | 92.33 (3.74) | 93.12 (3.44) | **85.39 (5.39)** |
| AdaptSegNet | ✓ | 82.61 (3.11) | 83.42 (3.18) | 87.02 (3.73) | 89.28 (4.95) | 89.80(3.88) | 90.60 (3.67) | 75.36 (7.34) |
| SHOT | | 89.68 (1.97) | 90.06 (2.03) | 91.01 (1.90) | 91.40 (3.06) | 90.56 (2.98) | 91.45 (3.26) | 81.67 (6.17) |
| AdaMI | | 90.65 (1.90) | 90.23 (2.07) | 92.60 (1.76) | 94.11 (3.25) | 93.55 (3.27) | 94.46 (3.00) | 82.10 (5.85) |
| SS-SFDA | | 87.04 (2.43) | 88.83 (2.35) | 89.90 (2.77) | 93.10 (3.63) | 92.37 (4.05) | 92.29 (3.93) | 68.39 (7.52) |
| Ours | | 90.49 (1.95) | **90.37 (2.02)** | **92.43 (1.89)** | **94.21 (2.94)** | **93.69 (2.89)** | **94.65 (2.78)** | 83.18 (5.85) |
| Oracle | | 91.20 (0.76) | 90.89 (1.15) | 91.93 (0.91) | 93.23 (1.19) | 92.29 (1.00) | 94.89 (1.04) | 88.22 (2.80) |

Table 3: Results (mIoU with standard deviation) in terms of teeth category on ATE.

## 3.5. Visualization

We demonstrate the superiority of our MATS with case visualization in Fig. 4. We focus on the contrast in the abnormal areas . It can be observed that the baseline methods tend to make mistakes in the three cases: (1) fail to recognize tooth-gingiva boundaries due to excess gingiva caused by Dentural Diastema; (2) fail to recognize complete molars due to Teeth Defects; (3) fail to segment neighboring teeth apart due to Abnormal Eruption. In contrast, our MATS can produce accurate predictions on these difficult samples.

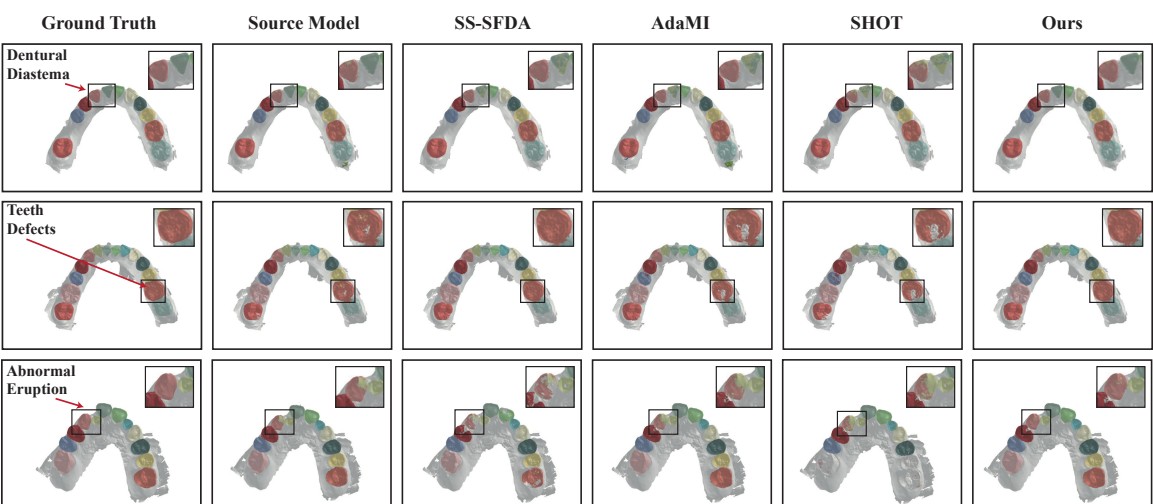

Figure 4: Visualization cases of segmentation. The "black box" indicates abnormal areas.

## 4. Conclusion

In this paper, we focus on solving two main challenges in practical tooth segmentation scenarios: expensive annotations and privacy concerns. Our method aims to build a robust tooth segmentation model based on 3D point cloud with only the well-trained source model and the target domain data so as to enable cross-center 3D tooth segmentation in real-world dental clinics. We achieve this goal by fully exploiting the tooth characteristics and designing three tooth prior-based modules. Extensive experiments were conducted on two self-collected datasets. The state-of-the-art performance demonstrated our superiority against existing SFDA and DA solutions.

## Acknowledgments

This work is supported by the National Natural Science Foundation of China (Grant No. 62106222), the Natural Science Foundation of Zhejiang Province, China(Grant No. LZ23F020008) and the Zhejiang University-Angelalign Inc. R&D Center for Intelligent Healthcare.

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

## Appendix A. Method

We provide a detailed version of our method for your information.

### A.1. Tooth-level Feature Prototypes Generation and Alignment

In the tooth segmentation task, tooth-level features exhibit similar patterns for two reasons. Firstly, for each tooth category, its appearance and frequency (usually once if not missing or hyperdontia) are quite consistent across tooth models, whether for patients in the source or target centers. Secondly, the spatial relationships among different tooth categories are determined, for example, the central incisors shall be near the incisors for most cases, leading to similar global features across IOSs. We, therefore, consider tooth-level features to be transferable across domains. Though source data is unavailable, we can instead retain tooth-level feature prototypes from the source domain. The generation operation of each feature prototype is as follows:

$$\mathcal{P}_k = \mathbb{E}_{x_s \in \mathcal{X}_s} \frac{\sum_i g_s(x_s^i) \mathbb{K}[y_s^i = k]}{\sum_i \mathbb{K}[y_s^i = k]}, \tag{4}$$

where $\mathcal{P}_k$ denotes the feature prototype for $k$-th tooth class, $\mathbb{K}[\cdot]$ is the indicator function, and $g_s(x_s^i)$ is the feature corresponding to the $i$-th point.

In the target domain, we measure the similarity between its features and generated prototypes for each point, and assign its class corresponding to the closest prototype. The distance is measured with the cosine similarity:

$$\phi(g_t(x_t^i), \mathcal{P}_k) = \frac{g_t(x_t^i) \cdot \mathcal{P}_k}{||g_t(x_t^i)|| \ ||\mathcal{P}_k||}. \tag{5}$$

For effective alignment, we expect the features of each point to be sufficiently similar to the closest prototype while away from the other prototypes. It means the similarity vectors should be close to one-hot encodings. Thus, we employ a self-entropy loss on similarity vectors,

$$\mathcal{L}_{se}(f_t, \mathcal{X}_t, \mathcal{P}_k) = -\mathbb{E}_{x_t \in \mathcal{X}_t} \sum_{i=1}^{n_t} \sum_{k=1}^{K} \phi(g_t(x_t^i), \mathcal{P}_k) \log \phi(g_t(x_t^i), \mathcal{P}_k). \tag{6}$$

Moreover, we observed that features of some points might be comparably close to multiple prototypes in experiments. For instance, a similarity vector like [0.33, 0.31, 0.35, 0.01]. This is likely because the prototypes of different teeth are close to each other. Take an instance from Fig. 7(A), prototypes of tooth 14 and tooth 15 have a similarity of 0.94. Obviously, it is troubled to align target features to these prototypes. Inspired by the concept of knowledge distillation (Hinton et al., 2015), we define these feature prototypes to be "over soft". In particular, we propose to "harder" the prototypes in the feature space by adding a softmax layer with low temperature on $\mathcal{P}_k$ in Eq. (4):

$$\mathcal{P}_k' = \frac{\exp(\mathcal{P}_k/\mathcal{T})}{\sum_k^K \exp(\mathcal{P}_k/\mathcal{T})}, \tag{7}$$

where $\mathcal{T}$ is the temperature. Correspondingly, the definition of $\mathcal{L}_{se}$ is changed as $\mathcal{L}_{se}(f_t; \mathcal{X}_t, \mathcal{P}_k')$. Detailed discussion on the effectiveness of $\mathcal{T}$ can be found in the experiments.

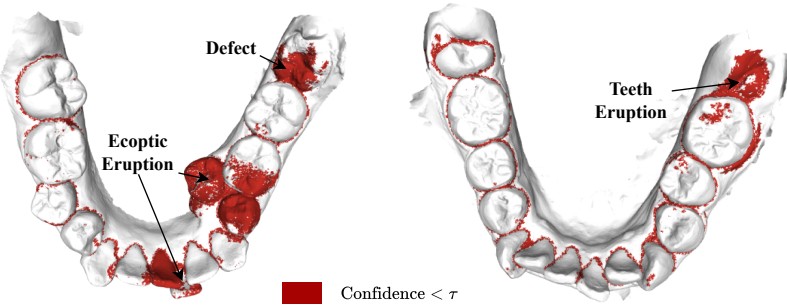

Figure 5: Visualization of confidence map generated by the source model. "Red" represents points with a confidence lower than the threshold $\tau = 0.9$ here, and "White" otherwise. Obviously, the low confidence points are concentrated in those areas where the dental diseases occur.

## A.2. Progressive Pseudo Label Self-training

In practice, the distribution difference between the target and source centers is mainly caused by teeth abnormalities. As we know, abnormalities always occur in an area within a dental model. For example, dentural diastema commonly appears as a gap in the anterior region. In our preliminary experiments, we have observed that points in the abnormal area are distinct from those in the source domain, while the rest points away from those abnormal areas tend to be similar to the source domain, as illustrated in Fig. 5. In this case, we make an assumption that the well-trained source model is more confident and accurate in predictions on similar points, while tends to produce wrong predictions on distinct points.

To efficiently adapt to the target center, we devise a progressive self-training strategy based on pseudo-labels. We propose to exclude a set of distinct points in the early training stage to encourage the model to focus on similar points first. In each training step, we determine the similar points according to the confidence level in model predictions. We then generate pseudo labels on these similar points for self-training. The generation process is as follows:

$$\tilde{y}_t^i = k, \quad \text{if} \quad \delta_k(f_t(x_t^i)) \geq \tau, \tag{8}$$

where $\delta_k(f_t(x_t^i))$ denotes the confidence level that the $i$-th point belongs to the $k$-th class. $\tau$ is the threshold. The point is excluded if $\delta_k(f_t(x_t^i)) < \tau, \forall k \in K$.

It is worth noting that we use the real-time target model rather than the frozen source model to generate pseudo labels. Hence, as the model becomes more adapted to the target domain, more points with high confidence will be selected during training, as characterized by the progressive learning procedure in our method. The loss function on the generated pseudo labels is defined as:

$$\mathcal{L}_p(f_t; \mathcal{X}_t, \tilde{\mathcal{Y}}_t) = - \sum_{(x_t, \tilde{y}_t) \in (\mathcal{X}_t, \tilde{\mathcal{Y}}_t)} \sum_{i=1}^{n_t} \tilde{y}_t^i \log \delta(f_t(x_t^i)). \tag{9}$$

It is likely that the model might generate wrong labels with high confidence (Chen et al., 2019a). To alleviate this issue, we further evaluate the quality of pseudo labels by

the similarity between their features and the corresponding prototypes. Pseudo labels of high similarity are given larger weights. The loss function is re-weighted as:

$$\mathcal{L}_p(f_t; \mathcal{X}_t, \tilde{\mathcal{Y}}_t) = -\sum_{(x_t, \tilde{y}_t) \in (\mathcal{X}_t, \tilde{\mathcal{Y}}_t)} \sum_{i=1}^{n_t} \phi(g_t(x_t^i), \mathcal{P}_k')) \tilde{y}_t^i \log \delta(f_t(x_t^i)), \tag{10}$$

where $\phi$ is the similarity between $g_t(x_t^i)$ and $\mathcal{P}_k'$.

### A.3. Information Maximization with Tooth-ratio Priors

In our method, we also adopt the Information Maximization (Peng et al., 2020; Liang et al., 2020) loss to supervise the output pattern. The IM loss consists of an entropy loss to encourage confident prediction and a diversity loss to encourage global diversity. The two losses are defined as follows:

$$\mathcal{L}_{ent}(f_t; \mathcal{X}_t) = -\mathbb{E}_{x_t \in \mathcal{X}_t} \sum_{i=1}^{n_t} \delta(f_t(x_t^i)) \log \delta(f_t(x_t^i)), \tag{11}$$

$$\mathcal{L}_{div}(f_t; \mathcal{X}_t) = D_{KL}(\hat{p}, \frac{1}{K} \mathbf{1}_K) - \log K, \tag{12}$$

where $\delta(f_t(x_t^i))$ is the K-dimensional probabilistic prediction of the $i$-th point in target dental model $x_t$ after softmax, $D_{KL}$ is the Kullback–Leibler divergence, and $\hat{p} = \mathbb{E}_{x_t \in \mathcal{X}_t}[\delta(f_t(x_t))]$ is the mean output embedding of the entire target domain. $\mathbf{1}_K$ is a K-dimension all-ones vector, illustrating another assumption made by the IM loss that different classes appear with equal probability.

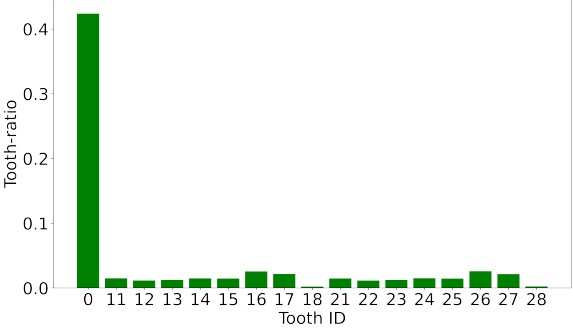

Figure 6: Visualization of tooth-ratio. The tooth-ratio represents the average proportion in all dental models for every tooth category. We report tooth-ratio for the gingiva (0) and the upper jaw (11-18, 21-28) in this figure.

However, such an assumption is not suitable for our tooth segmentation task. Based on our observations, in a universal dental model, the ratios between the teeth (and gingiva) tend to remain constant. As shown in Figure 6, appearance probabilities of molars (16, 17, 26, 27) are larger than the incisors (11, 21), and the probability of gingiva (0) is much larger than all teeth (11-18, 21-28). To cope with this issue, we propose to preserve these ratios in the source domain and transfer it to the target domain as prior knowledge (Vu et al., 2019;

Bateson et al., 2021). The generation of tooth-ratio priors for each class is as follows:

$$\mathcal{R}_s[k] = \frac{1}{|\mathcal{Y}_s|} \sum_{(x_s, y_s) \in (\mathcal{X}_s, \mathcal{Y}_s)} \frac{1}{|n_s|} \sum_{i=1}^{n_s} \mathbb{K}[y_s^i = k], \tag{13}$$

where $\mathbb{K}[y_s^i = k]$ is the indicator function. $|\mathcal{Y}_s|$ represents the number of dental models in the source domain. To further illustrate that our tooth-ratio prior is consistent across datasets and not change due to gender, age, etc, we report the tooth ratios in terms of gender in Tab. 4. It can be observed that the tooth ratio changes slightly due to the change of gender. Compared to the tooth-ratio on the whole dataset, the variation of the two subsets is also trivial.

| Tooth label | 0 | 11 | 12 | 13 | 14 | 15 | 16 | 17 | 18 |
|---|---|---|---|---|---|---|---|---|---|
| tooth ratio (female) | 0.4293 | 0.0197 | 0.0137 | 0.0153 | 0.01811 | 0.0169 | 0.0277 | 0.0249 | 0.0037 |
| tooth ratio (male) | 0.4401 | 0.0167 | 0.0134 | 0.014 | 0.0175 | 0.0175 | 0.0266 | 0.0215 | 0.0048 |
| tooth ratio (all) | 0.4356 | 0.0182 | 0.0136 | 0.0147 | 0.0177 | 0.0171 | 0.0275 | 0.023 | 0.0044 |

Table 4: Tooth ratios in terms of gender and the whole dataset.

Accordingly, we define the IM loss with a revised diversity loss based on the tooth-ratio priors $\mathcal{R}_s$, which is defined as:

$$\mathcal{L}_{IM}(f_t; \mathcal{X}_t) = \mathcal{L}_{ent}(f_t; \mathcal{X}_t) + \alpha D_{KL}(\hat{p}, \mathcal{R}_s), \tag{14}$$

where $\alpha$ is a hyper-parameter. The final loss function for MATS is defined as:

$$\mathcal{L} = \mathcal{L}_{IM} + \beta \mathcal{L}_{se} + \gamma \mathcal{L}_p, \tag{15}$$

where $\beta$ and $\gamma$ are weighting coefficient hyper-parameters.

## Appendix B. Dataset Description

### B.1. Dataset AbnTeeth

**Dental abnormalities.** Visualizations of the three dental abnormalities are presented in Fig. 7: (1) **Abnormal Teeth Eruption (ATE)**, describes a tooth erupting through the improper position. It causes problems in the alignment of its adjacent teeth. (2) **Dentural Diastema (DD)** describes the visible gap between teeth, resulting in the non-adjacent dentition. (3) **Teeth Defects (TD)** indicates the abnormality of tooth shape and structure, resulting in abnormal occlusion and adjacent relationship (Proffit et al., 2006). To further illustrate the domain gap caused by dental abnormalities, we present the statistical difference of features generated by the source model in Table 6.

| Disease | Total | Source (normal) | Target (abnormal) |
|---|---|---|---|
| Abnormal Eruption | 1278 | 655 | 623 |
| Dentural Diastema | 694 | 349 | 345 |
| Teeth Defects | 677 | 356 | 321 |

Table 5: The number of data samples in six domains.

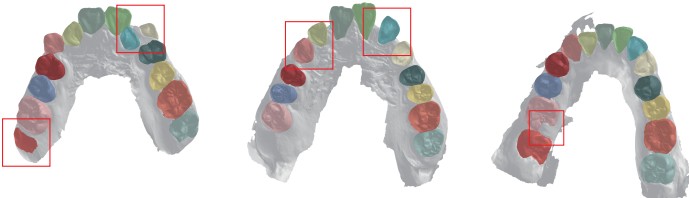

Figure 7: More visualizations on teeth abnormalities. The "red box" indicates abnormal areas. Left: Abnormal Tooth Eruption in improper location. Mid: Dentural Diastema (visible gaps between teeth). Right: Teeth Defects (incomplete molar).

| dim. | 1 | 2 | 3 | 4 | 5 | 6 | 7 | 8 |
|------|---|---|---|---|---|---|---|---|
| s.d. | -7.90 (23.32) | -7.58 (19.10) | -7.75 (20.60) | -8.86 (26.61) | -9.39 (37.78) | -7.81(36.23) | -7.05 (22.84) | -7.22 (15.92) |
| t.d. | -10.13 (22.55) | -9.70 (19.52) | -9.51 (20.28) | -10.73 (26.28) | -11.26 (37.51) | -9.97 (36.76) | -9.12 (22.18) | -8.83 (16.39) |
| dim. | 9 | 10 | 11 | 12 | 13 | 14 | 15 | 16 |
| s.d. | -7.92 (22.47) | -9.08 (29.62) | -9.18 (20.66) | -9.13 (27.21) | -9.07 (32.67) | -7.99 (35.10) | -7.72 (27.42) | -8.80 (32.70) |
| t.d. | -10.13 (22.02) | -11.02 (29.18) | -11.11 (20.38) | -10.75 (25.46) | -10.67 (30.82) | -9.91 (34.50) | -9.74 (27.39) | -10.27 (29.77) |
| dim. | 17 | 18 | 19 | 20 | 21 | 22 | 23 | 24 |
| s.d. | -17.58 (35.60) | -18.50 (41.47) | -19.15 (36.48) | -19.42 (49.25) | -19.69 (55.12) | -17.51 (49.93) | -16.43 (40.76) | -15.28 (31.25) |
| t.d. | -12.07 (28.36) | -12.68 (33.35) | -13.31 (29.39) | -13.33 (38.90) | -13.45 (43.10) | -11.77 (40.16) | -11.03 (32.10) | -11.04 (24.81) |
| dim. | 25 | 26 | 27 | 28 | 29 | 30 | 31 | 32 |
| s.d. | -17.17 (33.16) | -16.34 (33.08) | -16.32 (35.23) | -17.33 (43.23) | -19.36 (58.89) | -17.90 (56.44) | -16.75 (41.70) | -14.91 (23.52) |
| t.d. | -11.69 (26.24) | -10.68 (25.35) | -10.79 (27.15) | -11.60 (33.48) | -13.17 (47.38) | -12.00 (48.62) | -11.03 (32.53) | -10.67 (18.41) |

Table 6: Statistics (mean and variance) in different feature dimension (dim.) between the source domain (s.d.) and the target domain (t.d.) on ATE.

## B.2. Dataset CrossTeeth

**CrossTeeth.** is collected from 4 real-world clinical centers in different cities, which consists of 2,489 3D IOSs in total. We also report the statistics of gender and age of patients, with results in Tab. 7.

| Center | # of Data | Gender (female: male) | Age (avg.) |
|--------|-----------|-----------------------|------------|
| 1 | 500 | 326:174 | 20.33 |
| 2 | 499 | 383:117 | 26.54 |
| 3 | 490 | 435:65 | 26.9 |
| 4 | 500 | 326:174 | 23.54 |

Table 7: Statistics of CrossTeeth.

## Appendix C. Implementation Details

We employ a strong baseline model, DC-Net (Hao et al., 2022) as the backbone of the source and target model, which is an improved version of DGCNN (Wang et al., 2019). All other settings are kept the same as DC-Net. In the adaptation phase, we sample $n_s = n_t = 10,000$ points uniformly from each dental model. The target model is trained with AdamW optimizer with a learning rate of $lr = 1e - 6$ and a decay weight of $1e - 3$. The batch size is set to 2. Our method is trained with 100 epochs on a single RTX 3090 Nvidia GPU. The hyper-parameters, $\alpha$, $\beta$ and $\gamma$ are determined to be 1, 0.5, and 0.2. The threshold $\tau$ for the pseudo label is 0.9, and the temperature $\mathcal{T}$ for prototype separation is set to be 0.2. The hyper-parameters are obtained by traversing potential values on DD, and they are utilized for other tasks. Both $x_s$ and $x_t$ are point clouds with $n_s = n_t = 10,000$ points sampled from the original IOS mesh faces, and each point is associated with a 15-dimensional input feature including the 3D coordinates, normals and face shape descriptor following the settings in (Hao et al., 2022). The labels of $y_t^i$ are the same as $y_s^i$, where $y^i \in \{0; 11 - 18; 21 - 28; 31 - 38; 41 - 48\}$ denotes the gingiva and FDI notations for 32 permanent teeth.

## Appendix D. Ablation study

### D.1. Ablation study on the hyper-parameters

We investigate the sensitivity of the confidence threshold $\tau$, and the temperature $\mathcal{T}$. The results are presented in Figure 8 in terms of three metrics. We ranged $\tau$ from 70% to 99% and evaluated our model, results are provided in Tab.

The hyper-parameters are obtained by traversing potential values on DD and are utilized for other tasks. Ablation studies of hyper-parameters $\alpha$, $\beta$ and $\gamma$ are presented in Table 9, Table 10 and Table 11. We finally determined hyper-parameters, $\alpha$, $\beta$ and $\gamma$ to be 1, 0.5, and 0.2.

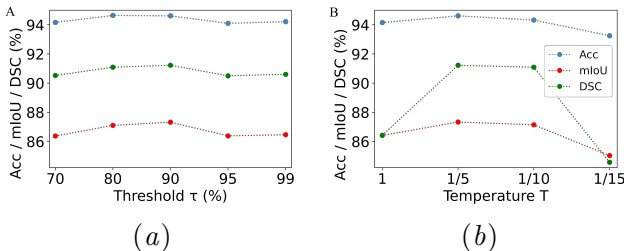

Figure 8: Ablation study of $\tau$ (left) and $\mathcal{T}$ (right).

|      | Acc(%) | mIoU(%) | DSC(%) |
|------|--------|---------|--------|
| 0.7  | 94.16  | 86.37   | 90.52  |
| 0.8  | 94.64  | 87.1    | 91.09  |
| 0.9  | 94.61  | 87.32   | 91.22  |
| 0.95 | 94.1   | 86.38   | 90.5   |
| 0.99 | 94.21  | 86.46   | 90.6   |

Table 8: Ablation study of $\tau$.

## D.2. Ablation Study on the losses

We further investigate the effectiveness of $\mathcal{L}_{IM}, \mathcal{L}_{se}$ and $\mathcal{L}_p$ on the ATE and TD transfer tasks. The results are shown in Table 12 and Table 13.

| $\alpha$ | mIoU (%) | DSC (%) | Acc (%) |
|------|----------|---------|---------|
| 0.2  | 84.95    | 89.54   | 93.36   |
| 0.5  | 85.05    | 89.59   | 93.41   |
| 1    | 87.32    | 91.22   | 94.61   |
| 2    | 85.00    | 89.50   | 93.46   |
| 5    | 85.22    | 89.71   | 93.55   |

Table 9: Ablation study of hyper-parameter $\alpha$.

| $\beta$ | mIoU (%) | DSC (%) | Acc (%) |
|---|---|---|---|
| 0.1 | 86.05 | 89.46 | 92.93 |
| 0.2 | 86.91 | 90.06 | 93.74 |
| 0.5 | 87.32 | 91.22 | 94.61 |
| 1 | 82.12 | 86.82 | 93.02 |
| 2 | 79.67 | 85.04 | 91.79 |

Table 10: Ablation study of hyper-parameter $\beta$.

| $\gamma$ | mIoU (%) | DSC (%) | Acc (%) |
|---|---|---|---|
| 0.05 | 84.28 | 88.34 | 94.01 |
| 0.1 | 83.54 | 87.88 | 93.58 |
| 0.2 | 87.32 | 91.22 | 94.61 |
| 0.5 | 86.09 | 89.19 | 93.86 |
| 1 | 85.97 | 88.36 | 94.04 |

Table 11: Ablation study of hyper-parameter $\gamma$.

## Appendix E. Comparisons on public datasets

We further evaluate our method on a public human segmentation dataset PointSegDA (Achituve et al., 2021). It has 4 subsets: adobe, faust, mit, and scape, which differ in point distribution, pose and, scanned humans. As shown in Tab .14, our method surpasses the DA and SFDA baselines by a large margin, demonstrating that our framework can be generalized to other similar tasks.

| $f_s$ | $\mathcal{L}_{IM}$ | $\mathcal{L}_{se}$ | $\mathcal{L}_p$ | mIoU (%) | DSC (%) | Acc (%) |
|---|---|---|---|---|---|---|
| ✓ | | | | 83.60 | 88.32 | 93.58 |
| ✓ | ✓ | | | 83.98 | 88.87 | 92.88 |
| ✓ | | ✓ | | 54.07 | 64.90 | 78.52 |
| ✓ | ✓ | ✓ | | 86.36 | 90.40 | 94.44 |
| ✓ | ✓ | | ✓ | 86.73 | 90.66 | 94.68 |
| ✓ | ✓ | ✓ | * | 86.20 | 90.32 | 94.38 |
| ✓ | ✓ | ✓ | ✓ | 87.32 | 91.10 | 94.98 |

Table 12: Ablation study of the losses on Abnormal Teeth Eruption. The "*" represents the unweighted version of $\mathcal{L}_p$.

| $f_s$ | $\mathcal{L}_{IM}$ | $\mathcal{L}_{se}$ | $\mathcal{L}_p$ | mIoU (%) | DSC (%) | Acc (%) |
|:---:|:---:|:---:|:---:|:---:|:---:|:---:|
| ✓ | | | | 80.38 | 86.02 | 91.62 |
| ✓ | ✓ | | | 83.20 | 88.43 | 92.10 |
| ✓ | | ✓ | | 35.92 | 46.30 | 63.50 |
| ✓ | ✓ | ✓ | | 84.71 | 89.37 | 93.35 |
| ✓ | ✓ | | ✓ | 84.96 | 89.55 | 93.42 |
| ✓ | ✓ | ✓ | * | 86.52 | 90.66 | 94.28 |
| ✓ | ✓ | ✓ | ✓ | 87.33 | 91.22 | 94.61 |

Table 13: Ablation study of the losses on Teeth Defects. The "*" represents the unweighted version of $\mathcal{L}_p$.

| Models | adobe to faust | faust to adobe | scape to faust | faust to scape | avg. |
|:---|:---:|:---:|:---:|:---:|:---:|
| Adapt-SegMap (Tsai et al., 2018b) | 44.2 | 60.1 | 65.3 | 70.1 | 59.93 |
| DefRec (Achituve et al., 2021) | 42.5 | 61.8 | **72.2** | 67.4 | 60.98 |
| SS-SFDA | 49.7 | 69.0 | 68.3 | 55.2 | 60.55 |
| Adapt-SegMap | **50.4** | **69.9** | 67.4 | **71.6** | **64.81** |

Table 14: Results of adaptation.

## Appendix F. Experiment on limited source data

To investigate the cases where the source model is not well-trained due to the limited data, we further design an experiment on different numbers of the source data. We randomly sampled 10%, 30%, 50%, and 70% of the numbers of data in the source domain to train the source model and keep all other settings unchanged. The results are reported (mIoU) in Tab. 15. It can be observed that with the decrease in training data, the performance of the source model declines in order. In all adaptation cases, our method brings steady improvements, which proves the robustness of our proposed technicals.

| Source Data | 10% | 30% | 50% | 70% | 100% |
|:---|:---:|:---:|:---:|:---:|:---:|
| source only | 73.09 | 81.01 | 81.09 | 82.32 | 83.76 |
| after adaptation | 79.00 | 85.38 | 85.94 | 86.89 | 87.90 |

Table 15: Results of limited source data.

## Appendix G. Tooth-level quantitative performance analysis

We further report the tooth-level quantitative performance on TD and DD in Table 16 and Table 17.

| Models | $\mathcal{D}_s$ | c-incisor | incisor | canine | $1^{st}$p-molar | $2^{nd}$p-molar | $1^{st}$molar | $2^{nd}$molar |
|---|---|---|---|---|---|---|---|---|
| source only | ✓ | 79.34 | 81.28 | 87.24 | 86.02 | 86.04 | 83.81 | 84.71 |
| MSL | ✓ | 88.53 | **89.69** | 91.20 | 90.28 | 89.70 | 90.38 | **88.36** |
| Adaptsegnet | ✓ | 83.56 | 81.85 | 86.35 | 86.56 | 84.04 | 88.74 | 84.87 |
| DAN | ✓ | 73.81 | 74.87 | 83.04 | 79.11 | 76.41 | 65.40 | 59.08 |
| PointDAN | ✓ | 84.59 | 86.28 | 89.86 | 86.42 | 83.81 | 85.16 | 80.98 |
| SHOT | | 82.29 | 82.26 | 87.52 | 83.92 | 80.57 | 79.69 | 76.57 |
| AdaMI | | 84.98 | 85.48 | 90.08 | 86.66 | 85.65 | 86.80 | 85.07 |
| SS-SFDA | | 88.28 | 89.25 | 91.69 | 90.81 | 90.42 | 91.08 | 87.01 |
| Ours | | **89.14** | 89.21 | **91.94** | **90.83** | **90.48** | **91.15** | 87.92 |
| oracle | | 89.98 | 89.97 | 92.21 | 93.09 | 90.80 | 90.57 | 88.08 |

Table 16: Results (mIoU) of adaptation on the abnormal teeth in terms of teeth category on TD.

| Models | $\mathcal{D}_s$ | c-incisor | incisor | canine | $1^{st}$p-molar | $2^{nd}$p-molar | $1^{st}$molar | $2^{nd}$molar |
|---|---|---|---|---|---|---|---|---|
| source only | ✓ | 87.73 | 86.01 | 91.43 | 85.84 | 84.47 | 86.05 | 81.39 |
| MSL | ✓ | 89.31 | 87.35 | 90.94 | 83.90 | 78.95 | 87.58 | 86.41 |
| Adaptsegnet | ✓ | 85.25 | 81.57 | 86.83 | 84.05 | 82.67 | 86.05 | 80.22 |
| DAN | ✓ | 72.37 | 74.23 | 85.96 | 84.64 | 80.67 | 79.98 | 55.62 |
| PointDAN | ✓ | 87.84 | 86.20 | 90.92 | 88.04 | 86.53 | 87.10 | 81.85 |
| SHOT | | 85.66 | 85.71 | 91.42 | 87.62 | 86.44 | 86.41 | 74.41 |
| AdaMI | | 90.16 | 87.38 | 91.59 | 85.96 | 84.89 | 87.29 | 83.50 |
| SS-SFDA | | 91.35 | **88.92** | 93.22 | 88.55 | **88.02** | 88.81 | 85.63 |
| Ours | | **91.53** | 88.89 | **93.28** | **88.65** | 87.76 | **89.64** | **86.89** |
| oracle | | 93.23 | 93.03 | 94.23 | 93.96 | 93.62 | 94.29 | 90.26 |

Table 17: Results (mIoU) of adaptation on the abnormal teeth in terms of teeth category on DD.

## Appendix H. Related Works

### H.1. Test-Time Adaptation

Test-Time Adaptation (TTA) aims to enable the adaptation of an existing model to new target data without having access to the source data. TTA can deal with dynamic domain shifts in the real world. TENT (Wang et al., 2020), the first Test-Time Adaptation (TTA) approach, proposes a simple yet effective entropy minimization method to optimize for test-time batch norm parameters without requiring any proxy task during training,

which is demonstrated for image classification and 2D semantic segmentation. Test-time Training (TTT) (Sun et al., 2020) updates model parameters in an online manner by applying a self-supervised proxy task on the test data. There are also other test-time training methods proposed to adapt a model with only online test data via self-supervised source model generation (Liu et al., 2021b), batch norm statistics (Lim et al.), test-time entropy minimization (Niu et al., 2022), Pseudo Labeling (Goyal et al., 2022), and feature alignment (Liu et al., 2021b). Compared to our setting, TTA can be considered a more strict SFDA setting where only the online unlabeled test data can be accessed only. It deals with the circumstances where distribution shifts arise at test time due to some natural factors, e.g., evolving road conditions (Gong et al., 2022), weather conditions (Bobu et al., 2018), or changes in demographics, users, and time periods (Koh et al., 2021). In our task, we focus on deploying the well-trained model to different new centers, which is not consistent with the TTA setting.

### H.2. 3D Tooth Segmentation

Recently, deep learning-based methods have achieved great performance in 3D tooth segmentation tasks. The pioneering work (Xu et al., 2018b) extracts predefined features which are subsequently processed by regular convolutional networks. The performance is significantly boosted by specific architecture designs that directly consume IOS meshes or point clouds, such as MeshSegNet (Lian et al., 2020), DC-Net (Hao et al., 2022), TSegNet (Cui et al., 2021), Mask-MCNet (Zanjani et al., 2021) etc. However, these methods usually rely on large-scale centralized annotated data, suffering from privacy concerns and distributional shift issues when deployed to multiple medical institutions. Some recent works explore unsupervised (He et al., 2021) or weakly-supervised tooth segmentation (Qiu et al., 2022). Though they can mitigate the time-consuming annotation issue, the privacy-preserving adaptation of trained models to new institutions with different oral diseases, which usually happens in practice, remains an unsolved problem.

### H.3. Source-Free Domain Adaptation

Source-Free Domain Adaptation (SFDA) extends unsupervised DA to more practical scenarios where source data is not available. Source Hypothesis Transfer (Liang et al., 2020) firstly exploits SFDA by self-supervised pseudo-labeling and then enforcing the model to comply with cluster assumption. Such techniques are utilized in many subsequent works (Li et al., 2021; Kothandaraman et al., 2021). Without using all source data, some works resort to domain alignment using the prototypes and statistical measurements of the source domain (Qiu et al., 2021; Stan and Rostami, 2021; Yeh et al., 2021). SFDA is also introduced to semantic segmentation task (Kundu et al., 2021; Liu et al., 2021a; Bateson et al., 2021; Kothandaraman et al., 2021; Yang et al., 2022; Qiu et al., 2021). From the algorithm perspective, this paper firstly investigates SFDA scenario for 3D meshes and point clouds, which has not been well exploited. The proposed algorithm aims to achieve robust 3D tooth segmentation across centers with different data distributions caused by dental abnormalities.

