# OpenReview forum: "Model Adaptive Tooth Segmentation"
_MIDL.io/2023/Conference — MIDL 2023 Oral_

### Official Review · Reviewer_Aswn · 2023-01-29

**Confidence:** 5
**Preliminary Rating:** 4
**Recommendation:** Poster

**Summary:**

The paper brings a method based on self-training to retrain a supervised method considering pseudo-labels in a self-supervised way. The supervised method is trained on data and labels from a source medical center, and the proposed self-supervised method is retrained on data from a target center.

The authors gathered two data sets: one contains no abnormalities and another one with them. This was mainly important to raise the performance of the proposed method.


**Strengths:**

The subject of the paper is pretty novel and relevant to the community. Annotation, especially segmentation, is always a cumbersome process. Ways to overcome this process are always welcome.

The paper shows a robust methodology to evaluate the proposed method considering two data sets (one containing tooth abnormalities and another without them) pursuing the goal of proving the performance of the proposed method in contrast with other baselines.

**Weaknesses:**

The arguments used to ground the proposal seem to be unstructured and weak. For example:

- "Automatic 3-dimensional tooth segmentation on intraoral scans (IOS) plays a pivotal role in computer-aided orthodontic treatments" - are you sure that it is always true? Is it worth spending more on all the treatment evaluations with IOSs? or would it be preferable to use 2D panoramic radiographs?

- "(1) the data distribution shifts between existing and new centers, (2) the data in the existing center is usually not allowed to share while annotating additional data in the new center is time-consuming and expensive."; in the introduction, the authors reinforce the strength of these arguments calling the IOS an "indispensable prerequisite" - these arguments are understandable. However, my question is how much they are true. I'm curious to know how many IOSs (per year) a center transfers to new medical centers to justify the proposal of the self-supervised method.

The critical part here is not the proposed method but the arguments to substantiate it.

Although most of the manuscript allows us to follow the main ideas, many parts need to be revised and rewritten due to confusing paragraphs and tense verb issues.









**Deanonymize Review:**

no

**Detailed Comments:**

I kindly ask the authors to clarify some points:

- The sub-index of some loss in the text, for example, "src" does not find references in Fig. 1. Please, try to revise the terms and variables in the text to meet the ones in the figure.

- "These modules inherit the merits of prevailing domain alignment techniques while revamping them to perform robust tooth segmentation" - what do you mean here? Is it possible to detail this idea in the text?

- After Eq. (2), the authors say, "For domain alignment, we expect the features of each point to be sufficiently similar to the closest prototype while away from the other prototypes, which means the similarity vectors should be close to one-hot encodings.". What if they (features) are not? what does it impact the proposed method?

- Eq. (5) is referred right after Eq. (2). Please, try to reorganize this part.

- In Section 2.3, the authors mention the existence of a threshold. What is the impact of this threshold on the choice of the pseudo-labels and, consequently, the performance of the self-supervised part of the method? why isn't this threshold in the ablation study?

- The use of 3 different metrics sometimes confuses the reader, mainly because the authors did not use the three metrics to carry out the performance analyses. For segmentation, DSC or mIoU are the most attractive ones.

- Why the ablation study is coming after the overall evaluation? I suggest swapping these sections.

- "For instance, if the abnormal eruption occurred near the Incisor, we only record the performance on the Incisor and exclude the performance on the other teeth." - why?


**Paper Type:**

validation/application paper

**Questions To Address In The Rebuttal:**

I kindly ask the authors to clarify some points:

- The sub-index of some loss in the text, for example, "src" does not find references in Fig. 1. Please, try to revise the terms and variables in the text to meet the ones in the figure.

- "These modules inherit the merits of prevailing domain alignment techniques while revamping them to perform robust tooth segmentation" - what do you mean here? Is it possible to detail this idea in the text?

- After Eq. (2), the authors say, "For domain alignment, we expect the features of each point to be sufficiently similar to the closest prototype while away from the other prototypes, which means the similarity vectors should be close to one-hot encodings.". What if they (features) are not? what does it impact the proposed method?

- Eq. (5) is referred right after Eq. (2). Please, try to reorganize this part.

- In Section 2.3, the authors mention the existence of a threshold. What is the impact of this threshold on the choice of the pseudo-labels and, consequently, the performance of the self-supervised part of the method? why isn't this threshold in the ablation study?

- The use of 3 different metrics sometimes confuses the reader, mainly because the authors did not use the three metrics to carry out the performance analyses. For segmentation, DSC or mIoU are the most attractive ones.

- Why the ablation study is coming after the overall evaluation? I suggest swapping these sections.

- "For instance, if the abnormal eruption occurred near the Incisor, we only record the performance on the Incisor and exclude the performance on the other teeth." - why?

---

### Official Review · Reviewer_kvWz · 2023-02-04

**Confidence:** 4
**Preliminary Rating:** 5
**Recommendation:** Oral, Poster

**Summary:**

This paper aimed to address the domain transfer problem in tooth segmentation. The proposed method relies on the source model and target data (no label). The novelty of the method is the use of tooth-level prototype, tooth-ratio prior, and pseudo label generation. This differs from conventional domain adaptation strategies since it leverages tooth characteristics and priors. The authors evaluated the method on two different datasets (consisting of several centers), showing better performance than existing methods.

**Strengths:**

+ Domain transfer is an important problem in medical imaging.
+ Method description and illustration are clear to follow.
+ Oracle performance of each center is provided as an upper bound reference.
+ The visualization of the abnormal area is very helpful.
+ The ablation study is extensive and convincing.

**Weaknesses:**

No major weakness is detected. Minor concerns are as follows:

- The tooth-ratio prior is assumed to be consistent across images (or datasets). It is expected to have statistics to support this point. Will this ratio change due to age, gender, or other factors (e.g., lost tooth, the additional tooth)? Analysis of the failure cases is suggested to include.

**Deanonymize Review:**

no

**Paper Type:**

methodological development

**Questions To Address In The Rebuttal:**

* This method presumes that the source model is well-trained. But what if the source model is not well-trained? Will the prototype and pseudo labels be robust? Any comments on improving the target model if the source model is not good (e.g., trained on very small data)?

---

### Official Review · Reviewer_oVbi · 2023-02-08

**Confidence:** 4
**Preliminary Rating:** 4
**Recommendation:** Oral

**Summary:**

In this paper, the authors proposed a source data-free domain adaptation method for tooth segmentation problem. The proposed method aims to tackle two problems: (1) domain gap between different centers, and (2) data privacy issue such that the training data from source center cannot be shared to target centers. Specifically the proposed approach incorporates a prototypical learning module, pseudo label learning and a loss function that considers the tooth prior information. Experimental results show that the proposed method could surpass the baseline methods as well as other UDA and SFDA methods.

**Strengths:**

- The paper is well-written and very clear in general.
- This paper studies an interesting problem for 3D tooth segmentation using point clouds under the setting of source data-free domain adaptation.
- The proposed tooth-prior loss is interesting and well-designed as it leverages the tooth ratio to further regularize the output.
- Details of the proposed method are explained well and experiments compared to other existing techniques are conducted in different settings and datasets.

**Weaknesses:**

There is one concern regarding the methodology of the proposed method.
For prototypical learning, the prototypes from source center are required. However, under the setting of source data-free domain adaption, it is not practical to have the access to the source-domain prototypes; typically, only the source model and target data are available. This may significantly limit the use of the proposed method in real clinical setting.

**Deanonymize Review:**

no

**Detailed Comments:**

- It would be better to include the literature of test-time adaption (TTA) in the literature review.
- The strategies of prototypical learning, self-training strategy and temperature scaling are similar to ProDA [1], a SOTA method for domain adaption. Please consider citing this paper.


[1] Zhang, P., Zhang, B., Zhang, T., Chen, D., Wang, Y., & Wen, F. (2021). Prototypical pseudo label denoising and target structure learning for domain adaptive semantic segmentation. In Proceedings of the IEEE/CVF conference on computer vision and pattern recognition (pp. 12414-12424).



**Paper Type:**

both

**Questions To Address In The Rebuttal:**

- In the rebuttal, I would like the authors to discuss why the source-domain prototypes can be accessible in source data-free domain adaption setting.
- Please include the literature review of test-time adaption, which also studies how to leverage the target data and source model.

---

### Meta-Review · Area_Chair_sUft · 2023-02-21

**Recommendation:** Accept (Oral)
**Confidence:** 5

**Metareview:**

The paper proposes a source data-free domain adaptation method for 3D tooth segmentation. The paper addresses a significant challenge of domain shift across different centers and data privacy. The proposed method is novel and well-motivated. The experiments are comprehensive and include ablation experiments and considering data with abnormalities.